# Promising Biomarkers of Radiation-Induced Lung Injury: A Review

**DOI:** 10.3390/biomedicines9091181

**Published:** 2021-09-08

**Authors:** Xinglong Liu, Chunlin Shao, Jiamei Fu

**Affiliations:** 1Institute of Radiation Medicine, Shanghai Medical College, Fudan University, Shanghai 200032, China; 20211140002@fudan.edu.cn; 2Department of Radiation Oncology, Shanghai Pulmonary Hospital, School of Medicine, Tongji University, Shanghai 200433, China

**Keywords:** radiation-induced lung injury, signaling pathways, animal models, biomarkers

## Abstract

Radiation-induced lung injury (RILI) is one of the main dose-limiting side effects in patients with thoracic cancer during radiotherapy. No reliable predictors or accurate risk models are currently available in clinical practice. Severe radiation pneumonitis (RP) or pulmonary fibrosis (PF) will reduce the quality of life, even when the anti-tumor treatment is effective for patients. Thus, precise prediction and early diagnosis of lung toxicity are critical to overcome this longstanding problem. This review summarizes the primary mechanisms and preclinical animal models of RILI reported in recent decades, and analyzes the most promising biomarkers for the early detection of lung complications. In general, ideal integrated models considering individual genetic susceptibility, clinical background parameters, and biological variations are encouraged to be built up, and more prospective investigations are still required to disclose the molecular mechanisms of RILI as well as to discover valuable intervention strategies.

## 1. Introduction

Radiation damage in normal lung tissue is inevitable during thoracic radiotherapy for either tumor control or palliative treatment. With the development of intensity modulation, motion management, and image guidance techniques, reduced toxicities and improved survival are achieved compared to older technologies [1,2,3]. However, radiation-induced lung injury (RILI) is still a great obstacle in the favorable prognosis of cancer patients, including those with radiation pneumonitis (RP) in the early stage and pulmonary fibrosis (PF) in the late stage. The incidence of pneumonitis was reported to be as high as 50% in lung cancer patients after conventional radiotherapy. With the use of intensity-modulated radiotherapy (IMRT), a significant reduction of RP risk was observed compared to three-dimensional conformal radiotherapy (3DCRT) [4]. IMRT was associated with less ≥grade 3 pneumonitis (7.9% vs. 3.5%, *p* = 0.039) in the secondary analysis of the RTOG 0617 clinical trial [5]. Nevertheless, the probability of symptomatic pneumonitis ranged from 9.4% to 28% despite the use of advanced technology of stereotactic body radiation therapy (SBRT) [6,7]. If extensive lung fibrosis occurs, therapeutics of antibiotics and corticosteroids are limited and there is no effective clinical treatment, which will seriously affect patients’ survival and quality of life [8]. Although combination treatments of cancers have become increasingly popular in current clinical practice, the complications of interstitial lung disease remain unclear. Radiation recall pneumonitis, where an acute or severe pneumonitis is triggered in previously irradiated regions after comprehensive therapies with chemotherapy or immunotherapy, has recently gained attention [9,10].

Many efforts have been made to identify key factors in the development of RILI. Based on a large amount of data from patients undergoing radiotherapy, some risk factors associated with lung injury have been identified. Clinically, treatment-related factors include fractioned dose prescription, dosimetric parameters, and systemic therapies [11]; tumor-related factors include tumor location, tumor volume, histology, and staging; patient-related factors include age, sex, smoking history, comorbid conditions, and intrinsic genetic phenotypes [12,13]. Clinicians endeavor to reduce the probability of adverse effects as well as control tumors through individual treatment modalities. However, the latent period of the occurrence of RILI is rather uncertain; it mostly occurs within weeks after radiotherapy, but can happen in a few months later. Since the clinical symptoms or imaging changes identifying toxicity do not appear in the early stage, ideal biomarkers are crucial for early diagnosis and intervention in order to prevent lung complications. Previous studies have discovered complicated intercellular interactions, signaling pathways, and dynamic cytokine cascades throughout the RILI process [14]. Nevertheless, no biological factors have been found to dynamically monitor RILI in cancer patients, and no reliable early predictors or accurate risk models are currently available in clinical practice.

This paper reviews newly discovered molecular mechanisms underlying RILI and assesses several commonly used preclinical animal models of RILI. We also summarized some valuable factors for predicting RILI, including crucial cells, cytokines, proteins, microRNAs, and genetic characteristics of individuals. It is important to establish an ideal model that can incorporate individual genetic susceptibility, clinical background parameters, and biological variations. Useful predictors will be beneficial in prevention and early diagnosis of lung toxicity after radiotherapy in thoracic cancer patients.

## 2. Molecular Mechanisms of Radiation-Induced Lung Injury

The progression of RILI generally consists of three highly overlapping and coordinated stages: acute radiation responses, radiation pneumonitis, and pulmonary fibrosis. Complicated intercellular crosstalk, various cascade reactions of pro-inflammation and anti-inflammation, and microenvironmental remodeling have been demonstrated in the pathogenesis of RILI [15,16]. In Figure 1, the vital immune cell types, critical signaling factors, and pathways have been detailed. In particular, the roles of T-cell subsets and macrophages in promoting pneumonitis and lung fibrosis, and the newly discovered mechanisms of epithelial-mesenchymal transformation (EMT) and extracellular matrix (ECM) in fibrogenesis and scar formation have been emphasized.

During acute responses after radiation exposure, reactive oxygen species (ROS) are generated immediately from the hydrolyzed water in the lung tissue. Free radicals and lipid peroxidation products can interact with molecules of proteins, the nucleus, cellular components, or ECM simultaneously [17]. DNA single-strand breaks (SSBs) and double-strand breaks (DSBs) can be induced in alveolar epithelial cells (AECs) and vascular endothelial cells directly or indirectly [16]. Both unrepaired DNA damage and uncontrollable oxidative stress result in the injury, death, or apoptosis of vital cells, which are sensed by a series of inflammatory cells including macrophages, lymphocytes, neutrophils, and monocytes. Thereafter, a series of cytokines, such as tumor necrosis factor (TNF)-α, interleukin (IL)-1β, IL-6, and transforming growth factor (TGF-β1), are released [18]. Radiation-induced oxidative stress also activates the mitogen-activated protein kinases (MAPKs) and nuclear factor (NF)-κB pathways by regulating protein kinases and protein phosphatase receptors, which enhance the production of cytokines in a feedback mechanism [19]. In addition, oxygen consumption leads to tissue hypoxia, which promotes endothelial cell proliferation via vascular endothelial growth factor (VEGF) generation [20,21]. A recent report revealed that the dynamic changes in mitochondrial genes and local oxidative stress are involved in the overall development of RILI [22].

AEC I and AEC II constitute the human alveolar epithelium. AEC II can differentiate to AEC I when the level of AEC I decreases, which is also relevant to the tissue repair procedure. A key role of epithelial senescence and damage has recently been highlighted in lung pathology [23]. The human oncoprotein MDM2 initiates DNA replication in lung progenitor cells and promotes epithelial repair by activating the Akt signaling pathway [24]. Previous studies have shown that damaged or dying cells can release damage-associated molecular patterns (DAMPs), which can activate signal transducer and activator of transcription (STAT) or signal transduction proteins, such as intercellular adhesion molecule 1 (ICAM1) and vascular cell adhesion molecule 1 (VCAM1) [25]. Toll-like receptor 4 (TLR4) is a vital receptor of macrophages for recognizing DAMPs [26]. Furthermore, these molecules message various innate immune cells for tissue repair and regeneration, resulting in neutrophil, lymphocytic, and macrophage transmigration. Infiltration of inflammatory cells leads to the secretion and release of numerous cytokines via autocrine and paracrine pathways in the alveolar space, including pro-inflammatory cytokines such as TNF-α, IL-1β, IL-6, IL-17, IL-8, and anti-inflammatory cytokines such as IL-10, IL-4, IL-13, and TGF-β [27]. There is a balance between inflammation and restoration in the damaged region for tissue homeostasis. When this balance collapses, pneumonitis develops, consistent with immunological disorders and cytokine storms.

Macrophages are the primary mediators of the chronic inflammatory response and are divided into two phenotypes with different functions [28]. The M1 macrophages function as pro-inflammatory mediators by secreting cytokines and promoting ECM degradation via matrix metalloproteinase (MMP) production [29,30]. The M2 macrophages play an essential role in the pathogenesis of pre-fibrotic clusters in lung fibrosis through TGF-β secretion [31,32,33]. In addition to macrophages, lymphocytes are another prominent cell type that are closely related to severe lymphoid interstitial pneumonitis. Various T lymphocyte subsets are responsible for maintaining the balance of the immune system by different cytokines. T helper (Th) cells and regulatory T cells (Tregs) have been recognized in the development of RILI in numerous studies.

Th1 cells play an important role in suppressing fibroblast proliferation and reducing excessive collagen production through interferon (IFN)-γ secretion. Th1 cells also inhibit the proliferation of Th2 cells and stimulate the activation of M1 macrophages through cytokines such as macrophage colony-stimulating factor (M-CSF). Th2 cells contribute to fibroblast proliferation and collagen production by upregulating cytokines IL-4 and IL-13. Reduction of IL-4 levels can attenuate the infiltration of macrophages and lymphocytes [34]. The role of Th17 cells in RILI remains unclear. Animal studies of bleomycin-induced lung injury suggested that IL-17A-deficient mice showed attenuated lung fibrosis effects compared to wild-type mice. High levels of IL-17 have also been implicated in the pathology of pneumonitis [35,36]. Moreover, as a subset of CD4^+^ T cells, Tregs are involved in RILI development, and their anti- or pro-fibrotic effects are dependent on disease stages via complex regulatory networks, such as the promotion of EMT, the modulation of Th1/Th2 balance, and the suppression of Th17 responses [37,38].

During lung fibrosis, fibroblast recruitment and myofibroblast proliferation are crucial for ECM remodeling [39]. Contribution to the accumulation of fibroblasts and myofibroblasts is facilitated by cell proliferation in situ, EMT, and fibrocyte derivation from circulating bone marrow [40]. Among these, EMT plays a crucial role in tissue fibrosis, which is described as a consecutive process of epithelial cells losing their initial phenotypes, transforming to an intermediator, and finally exhibiting traits of mesenchymal fibroblast [41,42,43]. Substantial evidence has shown that the mechanism of EMT is associated with progressive organ fibrosis [44,45]. When the lung is exposed to irradiation, lung stem cells (LSCs) differentiate into AEC II that further differentiate into myofibroblasts and AEC I [46]. The ERK/GSK3β/Snail signaling pathway in AEC II is relevant to EMT in rat lung after irradiation [47]. A recent study found that granulocyte-macrophage colony-stimulating factor (GM-CSF) alleviated pulmonary fibrosis by reducing the incidence of EMT by inhibiting the production of inflammatory cytokines such as TGF-β, TNF-α, IL-1β, and IL-6. EMT could not only be directly induced by ionizing radiation, but can also result indirectly from TGF-β [48].

TGF-β is the most critical molecule involved in fibrosis, and its mechanism of action in radiation-induced pulmonary fibrosis is summarized in Figure 1. TGF-β partly functions in a pro-fibrotic role by binding to the transmembrane proteins of serine/threonine kinase to transduce signals. Multiple signaling cascades, including ERK/GSK3β/Snail, Smad/Snail, and PI3K/AKT/mTOR axis, are activated [49,50,51]. It has been demonstrated that TGF-β enhances the expression of Insulin-like growth factor binding protein (IGFBP7), which further activates the ERK signaling pathway to promote EMT of AEC II [52]. Other molecules, such as αvβ6 integrin, ROS, and vitronectin (VTN), were also observed to be involved in fibrogenesis by promoting the expression of TGF-β [53,54]. In general, TGF-β displays wide regulatory effects on ECM formation, such as triggering the proliferation and differentiation of fibroblasts into myofibroblasts, overproduction of collagen and fibers in myofibroblasts, and inhibition of collagen degradation. These phenomena result in excessive ECM deposition in the lung parenchyma. Irreversible scar formation ultimately leads to pulmonary fibrosis and loss of lung function.

## 3. Preclinical Animal Models of Radiation-Induced Lung Injury

Reliable and appropriate animal models, which can aptly mimic clinical outcomes in human beings, are critical to successful translational experiments, such as disclosing the underlying molecular mechanism, developing new prevention and treatment strategies, and making comparisons across different institutions. Animals used for the RILI model mainly include mice, rats, rabbits, dogs, swine, and non-human primates. Among different animal species, the timing and severity of lung injury after the same dose of radiation varies dramatically. Recently, owing to their similarities to human beings in anatomy, physiology, pathology, and genomics, mice are the most widely applied models, including different strains such as C57BL/6, CBA, C3H, BALB/c, LAF1, and CD2F1. In particular, the C57BL/6 strain has been increasingly used owing to its advantages of high radiosensitivity of pulmonary fibrosis and that the radiation responses are closely related to humans. Another commonly used mouse strain is C3H/He, which is highly suitable for the development of early-stage pneumonitis [55]. Considering the variant responses among different mouse strains, it is recommended that at least two genetic strains should be used to assess the molecular mechanisms, predictive biomarkers, and therapeutic agents [56].

In earlier studies, whole thorax or whole-body irradiation was mainly applied with a single high dose ranging from to 10–25 Gy, and doses exceeding 15 Gy were mostly used for C57BL/6 mice [56]. Although the severity of inflammation and fibrosis differed with irradiation dose, similar pathologies of disease progression were observed, including increase in alveolar septum thickness, inflammatory cell infiltration, fibroblast proliferation, and collagen deposition [57]. Sophisticated radiotherapy techniques are now widely being used in clinical practice. When SBRT is applied, non-irradiated lung tissue has a strong compensatory function [58]. Thus, localized irradiation targeting a sub-volume of mouse lung was reported to better mimic clinically accurate radiotherapy [59]. The radiation responses were related to the targeted site in the C3Hf/Kam female mouse lung. For the same volume of irradiation, the damage degree generated from the irradiated base was always greater than that from the irradiated apex [60]. A recent report indicated that fibrosis-relevant gene expression profiles were different in the 90 Gy SBRT-mimicking model compared to the 20 Gy conventional radiotherapy model [61]. Generally, a single dose or hypo-fractional dose in the whole lung is used for short-term experiments and rapid establishment of the RILI model, while focal irradiation, with a more precise dose, is more suitable for observing long-term changes in lung function.

Another key issue in animal models is the evaluation of lung toxicities. Classical histopathological diagnosis remains the standard for the qualitative analysis of the successful establishment of animal models. Using hematoxylin and eosin (H&E) and Masson’s trichrome staining, radiation-induced pneumonitis and fibrosis are assessed and scored according to the semi-quantitative analysis of parenchymal alveolitis and fibrosis formation in the lung tissue [62]. However, there are several disadvantages. First, the execution of mice in batches is required during specimen collection. Moreover, the dynamic development of the injury cannot be observed consistently, and individual variations in mice are poorly controlled. With technological advancements, small animal micro-CT can overcome the above shortcomings. Disease progression can be monitored in real-time without killing the animals. This innovative technique not only benefits the sample size but also provides high-resolution images with 3D reconstruction and visualization that are correlated with human patient data. Nonetheless, the current cost of the technique is too high for it to be popularized.

The lethality endpoint has been used as a primary criterion in numerous studies evaluating radioprotectors or radiomitigators, according to the Food and Drug Administration’s (FDA) ‘Animal Rule’ [63]. However, it has not been concluded whether severe pneumonitis and pulmonary fibrosis are significant causes of mortality. Some studies have shown that other complications, including pleural effusions and respiratory failure, were present during the late phase as observed in autopsy examinations [64,65]. It was assumed that a continuing cascade of various cytokines leads to lung damage, and the relative concentrations of these factors were positively correlated with the severity of lung toxicities. The detection of cytokines and enzymes is widely used because of their high sensitivity. TNF-α, IL-1, and IL-6 are commonly measured pro-inflammatory cytokines, while TGF-β and hydroxyproline are frequently examined fibrosis-relevant factors [66,67]. Although various temporal patterns of cytokines have been documented in different laboratories, they cannot replace assays of pulmonary pathophysiology for monitoring the occurrence of acute pneumonitis and later fibrosis. The exploration of more valuable noninvasive functional and imaging parameters is needed. Several studies have focused on identifying in vivo imaging probes to directly target progressive fibrosis for early diagnosis [68,69].

## 4. Potential Biomarkers for Monitoring Radiation-Induced Lung Injury

Biomarkers for the early prediction of lung toxicity are crucial for preventing disease progression and reducing patient mortality. The most commonly used methods to decrease the probability of radiation pneumonitis are based on the clinical dose-volume relationship [70,71]. However, even with acceptable dosimetric parameters, patients still face the threat of pulmonary complications. There are no reliable and accurate molecular factors that could dynamically indicate RILI in cancer patients, as many external and internal co-driving factors contribute to the development of lung toxicity. According to the molecular mechanisms discussed in Section 2, promising biological factors for RILI assessment are reviewed in Table 1 and Table 2.

### 4.1. Key Immune Cells

An increasing number of studies have suggested that inflammatory responses are closely associated with RILI. Especially in RP, the variation in the number of immune cells is a potential indicator (Table 1). It was found that CD4^+^ T lymphocyte levels were obviously lower in the early period of RP, and decreased lymphocyte count was associated with the severity of RP in lung cancer patients after thoracic radiotherapy, which was consistent with animal experiments [109]. Moreover, as T cells determine the specificity of immune responses in tissue inflammation, the quantity of CD4^+^ T cells and CD4^+^:CD8^+^ T cell ratio decreased significantly at the onset of RP compared with the no-RP group, and increased CD4^+^ T-cell quantity and reduced C-reactive protein (CRP) level were observed after effective steroid therapy [72,73]. Clinically, higher neutrophil-lymphocyte ratio (NLR) has been used as a biomarker of systemic inflammation, and it is relevant to the poor prognosis of cancer patients [77]. For stage III non-small-cell lung cancer (NSCLC) patients with radiological RP, NLR was used to predict subsequent progression to symptomatic RP, although their lymphocyte counts were not significantly different initially [76,77].

In addition to RP, a study revealed that the frequency of Th17 cells increased after irradiation, which was associated with PF in mice. The activation of the IL-6/TGF-β/IL-17 signaling pathway is involved in radiation-induced severe fibrosis [74,75]. These results suggest that T-cell differentiation and proliferation are altered dynamically during the development of RILI. With the development of single-cell RNA sequencing (scRNA-seq) technology, comprehensive human single-cell landscapes have been constructed, especially for lung immune and collagen-producing lung cells [110,111]. scRNA-seq, a central genome-wide sequencing method to portray cellular identities, is capable of exploring the differentiation procedure of key immune cells and distinguishing the source of important cytokines or proteins in RILI [112].

### 4.2. Cytokines and Proteins

As described in Table 1, cytokines and proteins play essential roles in the process of lung damage. Owing to the convenience of blood sample collection, blood testing has become the most promising method to identify RILI risk [113,114]. In this review, relevant molecular factors of lung toxicities were divided into four categories according to their functions, including inflammation-related factors, fibrosis-related factors, chemokines, and other proteins (Table 1).

Inflammation-related factors are involved in acute or chronic inflammatory responses. TNF-α is the main initiator of the pro-inflammatory cascade by activating the expression of transcription factors, intercellular adhesion molecules, and numerous acute phase proteins. TNF-α also promotes fibroblast growth and collagen deposition [115,116]. Overproduction of TNF-α after irradiation has been well documented to be correlated with early cell apoptosis and latent lung function damage [117]. Augmented levels of TNF-α in the plasma were observed after radiotherapy. Although TNF-α level is related to RP, it failed to be used as a predictor in a previous study [118]. As another vital pro-inflammatory cytokine in human diseases, IL-1β can promote the recruitment of inflammatory cells by inducing the expression of adhesion molecules on endothelial cells and enhancing chemokine release [78]. The level of IL-1β was significantly elevated after thoracic irradiation both in vivo and in vitro; this could trigger the production of TGF-β and IL-6, which are involved in fibroblast proliferation and interstitial cell infiltration [79].

IL-6 plays a critical role in regulating lymphocyte proliferation and differentiation and participates in acute phase responses and immune hematopoiesis. Overexpression of IL-6 was detected clinically in the earlier phase of lung toxicity, which is closely related to the development of RP [80,81]. Nonetheless, it is still controversial whether IL-6 is a predictor of RILI because of its non-specific effects on inflammation [82]. IL-10 functions as an anti-inflammatory cytokine by suppressing the production of pro-inflammatory cytokines, including IL-6, and inhibiting the capabilities of antigen-presenting cells. Persistent low levels of IL-10 were found in RP patients throughout the radiotherapy period, and varied IL-10 concentrations were detected among different RP scales [83]. Moreover, the level of another inflammatory factor, CRP, was found to be higher at the onset of RP than in the no-RP group in lung cancer patients [72]. CRP has a prognostic value in disease severity and is associated with adverse outcomes in patients with lung infections. CRP, combined with other relevant factors, is a potential predictor of radiation toxicity [119,120].

As the most important pro-fibrotic factor, TGF-β1 promotes the differentiation of fibroblasts into myofibroblasts and the synthesis of ECM proteins [51]. Its central role in fibrosis progression is summarized in Section 2. Numerous studies have confirmed that plasma TGF-β1 levels are significantly correlated with both pulmonary and non-pulmonary radiation toxicity in patients after radiotherapy [121,122,123]. It was validated that higher levels of TGF-β1 and lower baseline levels of IL-8 were significantly associated with an increased grade ≥ 2 risk of RILI in NSCLC patients treated with definitive radiotherapy [86]. IL-8, an anti-inflammatory cytokine that is associated with symptomatic inflammatory responses, induces lung fibrosis by augmenting the mesenchymal cell population and recruiting more activated macrophages [84]. In addition, IL-4 and IL-13 play key roles in the adaptive immune response, and are both Th2-derived pro-fibrotic cytokines. In rat models, IL-4 levels increased with time after irradiation, leading to type I and III collagen formation in fibroblasts [124]. Both IL-4 and IL-13 can promote fibrosis by upregulating the expression of cell surface β1 integrin and VCAM-1 and simulating M2 macrophage polarization [31,125].

Although excessive ECM deposition is crucial in lung fibrosis, ECM initially contributes to wound healing of damaged lung parenchyma by binding cytokines and growth factors, such as platelet-derived growth factors (PDGFs) [126]. PDGFs are released from the endothelium, macrophages, or platelets, which can accelerate tissue repair by stimulating cell proliferation and ECM synthesis [127]. PDGF receptor α (PDGFR-α) is transactivated by TGF-β1 and is involved in tissue fibrosis. High levels of PDGF/PDGFR have been observed in lung, liver, kidney, and heart fibrosis [128]. Moreover, endothelin-1 (ET-1) is present downstream of TGF-β1. On the one hand, ET-1 can inhibit the proliferation and migration of endothelial cells, while on the other hand it promotes the expression of fibrosis-associated genes, such as plasminogen activator inhibitor-1 (*PAI-1*) [87]. ET-1 inhibition prolonged the survival of patients with idiopathic pulmonary fibrosis (IPF) [88]. In irradiated mice, serum ET-1 levels were obviously increased in the early stage, potentially indicating dynamic changes in lung injury.

In addition, the TGF-β-related protein Krebs von den Lungen-6 (KL-6/MUC1) has chemotactic and anti-apoptotic effects on fibroblasts. The serum level of KL-6/MUC1 reflects the severity of interstitial lung disease associated with connective tissue disease [90,91]. KL-6 is produced by epithelial cells, especially AEC II, and is released from these damaged cells following irradiation. It was reported that serum KL-6 levels increased almost in line with the occurrence of grade ≥2 RP and decreased after steroid administration in NSCLC patients [89]. Moreover, surfactants secreted by type II pneumocytes are vital for maintaining alveolar structure and function. The increased club cell secretory protein/surfactant protein D (CCSP/SP-D) ratio in plasma was positively correlated with fibrosis development and was detected early after radiation exposure in a murine model [97,98]. SP-D functions in host defense and regulates immune responses and lung phospholipid levels [129]. Previous studies have suggested that circulating SP-D levels were elevated in RP patients, which may be a sensitive and useful biomarker for early RP prediction [99].

Chemokines are associated with the migration of immune cells in response to infection or inflammation [130]. Chemokine C-C motif ligand 2 (CCL2), also known as monocyte chemoattractant protein-1 (MCP-1), is a potent chemokine for monocytes and contributes to lung pneumonitis and fibrosis [95]. The CCL2 level in the plasma was significantly lower in patients with grade 2 RP [81,86]. Furthermore, using high-throughput detection methods, the bronchoalveolar lavage fluid (BALF) of NSCLC patients was analyzed. The results showed that several cytokines, including C-X-C motif chemokine ligand 1 (CXCL-1), PAI-1, and IFN-γ, were upregulated in grade ≥ 3 RP patients [93]. Among them, CXCL-1 is a remarkable neutrophil chemoattractant and is involved in angiogenesis and inflammation. PAI-1 inhibits the plasmin system by blocking fibrinolysis and degradation of the ECM, and plays an important role in the development of PF [92,94]. IFN-γ is a pleiotropic cytokine with antitumor, antiviral, antibacterial, pro-inflammatory, and antifibrotic properties [96]. IFN-γ secreted by Th1 cells can inhibit Th2 cell differentiation and Th2-derived cytokine expression (IL-4 and IL-13), which further attenuates fibrosis formation by restricting fibroblast proliferation and reducing excessive collagen production [131].

### 4.3. MicroRNAs

MicroRNAs (miRNAs) are single-stranded, highly conservative small noncoding RNAs [132] involved in the regulation of gene expression, transcription, translation, and epigenetic modification; miRNAs have been studied in various fields, including disease diagnostics and cancer therapeutics [133,134,135]. The relationship between miRNAs and radiosensitivity or radiotoxicity has been reported in recent years. MiR-18-5p was reported to be a potential target for improving radiosensitivity in lung cancer by regulating the ataxia telangiectasia mutated (*ATM*) gene and hypoxia-inducible factor 1 alpha (*HIF-1α*) gene [136]. Additionally, overexpression of miR-26b-5p inhibited the expression of activating transcription factor 2 (*ATF2*) gene, which resulted in enhanced radiosensitivity of A549 lung adenocarcinoma cells [137].

During the acute phase after irradiation, the lower levels of miR-21 were related to a higher incidence and grade of RILI in patients by increasing the expression of IL-6 and TNF-α [100]. Similarly, another key protective molecule, miR-140, protected normal lung tissue from fibrosis by blocking TGF-β1 signaling and inhibiting myofibroblast differentiation [101]. Decreased levels of circulating miR-29a-3p and miR-150-5p in secreted exosomes were correlated with the delivered RT dose, and miR-29a might be useful biomarkers for lung fibrosis after irradiation [138]. Prominent changes in systemic miRNA profiles were observed within the early period after radiation exposure; miR-34b-3p, -96-5p, and -802-5p were identified to be associated with the TGF-β/SMAD signaling in C57BL/6 mice [139]. MiRNAs are stable in tissues or plasma and can be easily detected. Therefore, it is worthwhile to explore the value of circulating miRNA signatures for monitoring RILI.

### 4.4. Genetic Characteristics

Clinically, there is great variability among cancer patients in response to radiotherapy. Therefore, it is possible that the genetic characteristics of individuals are significantly associated with RILI occurrence [140,141]. Single-nucleotide polymorphisms (SNPs) have been considered hotspots in the development of RILI. A previous review summarized several SNPs in DNA repair-, inflammation-, angiogenesis-, and stress response-related genes that were involved in the underlying mechanisms of RP [142]. New discoveries continue to enrich the contents of SNPs related to RILI, and the latest findings are reviewed in this paper in Table 2.

Among the DNA repair-related genes, the SNP rs1051772 of topoisomerase DNA binding II binding protein 1 (*TOPBP1*) gene was found to be associated with a decreased risk of RP in NSCLC patients through DNA replication checkpoint control and genomic stability maintenance [102,143], and SNP rs1801131 of the methylenetetrahydrofolate reductase (*MTHFR*) gene was reported to be statistically correlated with grade ≥2 RP among esophageal squamous cell carcinoma patients in the Chinese Han population [103]. Nei endonuclease VIII-like 1 (NEIL1) is a bifunctional enzyme in the base excision repair (BER) pathway. Genetic variants (rs4462560 and rs7402844) of *NEIL1* gene can serve as independent biomarkers for predicting RP in patients treated with thoracic radiotherapy through regulation of NEIL1 expression [104]. Moreover, the inflammation-related gene *IL4*: rs2243250 was validated as a potential predictor of severe injury due to its high correlation with grade ≥ 3 RP in lung cancer patients [13]. The variant of autophagy-related gene autophagy related 16 Like 2 (*ATG16L2*) could predict the RP risk and better prognosis in NSCLC patients treated with radiotherapy, either with or without chemotherapy [106].

For stress response-related genes, the association between the glutathione S-transferase-P1 (*GSTP1)* Ile105Val polymorphism and the risk of RP was observed in lung cancer patients of the Chinese population [144]. The *GSTP1* gene abundantly expresses glutathione S-transferase (GST), a detoxification enzyme that protects against radiation-induced oxidative damage [145]. In addition, homeodomain-interacting protein kinase 2 (HIPK2) is a member of the serine/threonine kinase family, and there is a clearly increased risk of RP in lung cancer patients with the CC genotype of *HIPK2*: rs2030712 after radiotherapy [12]. A team has systematically investigated a series of genes relevant to RP in lung cancer patients, and found that the SNP rs4665162 of integrin beta6 subunit (*ITGB6*) gene, rs1144393 in matrix metalloproteinase-1 (*MMP-1*) gene, and three SNPs in the phosphatidylinositol 3-kinase (PI3K)/AKT pathway (*PI3CA*: rs9838117, *AKT2*: rs33933140 and rs11880261) were associated with higher RP risk [105,107,108]. Among them, the PI3K/AKT pathway is an important downstream pathway of TGF-β involved in the pathogenesis of inflammation and fibrosis diseases [146,147]. Although the clinical application of the above potential markers of SNPs in genes requires further prospective studies, SNPs are still a potential strategy for selecting high-risk patients for radiotherapy toxicity.

### 4.5. Imaging Based Biomarkers and Others

As there is a latency between radiotherapy and symptomatic lung toxicity, changes in the lung can be detected by advanced imaging technologies. Some non-invasive biomarkers are under investigation for early diagnosis of RILI. For thoracic tumor patients after radiotherapy, studies showed that vascular damage led to the reduction of regional lung perfusion. By using two functional single-photon emission computed tomography (SPECT) probes, regional pulmonary perfusion and pulmonary cell death were measured in the same irradiated rat model. The results demonstrated that the decrease of perfused volume of the lung was closely associated with lung tissue damage [148,149]. Especially in patients with RP, perfusion reduction was found more serious in high dose regions than those without RP with the performance of SPECT/CT [150]. Besides lung perfusion, alterations in pulmonary vascular resistance and permeability are also potential indicators of lung injury. A whole body contrast enhanced dynamic near-infrared fluorescence imaging was recently carried out in a rat model to track lung vascular permeability after acute RILI [151].

In addition, pre-treatment [18F]-2-fluoro-2-deoxyglucose positron emission tomography (FDG PET) imaging has also been investigated to predict RILI in several studies. Petit et al. firstly evaluated the correlation between symptomatic RP and pre-treatment FDG PET evidence of pulmonary inflammation. This retrospective study found that the 95th percentile of the standard uptake value (SUV95) within the lungs was potentially a predictor of RP on multivariate analysis [152]. The value of SUV95 for assessing lung toxicity was similarly validated by the treatment modality of the SBRT technique [153]. Through quantitative assessment of FDG PET/CT imaging before and after radiation therapy in stage III NSCLC patients, a pilot study suggested that global lung parenchymal glycolysis and lung parenchymal SUVmean may serve as potentially useful biomarkers to lung inflammation after thoracic radiation therapy [154]. Moreover, based on pretreatment planning CT in patients after SBRT delivery, the radiomic predictive model of lung volume irradiated with more than 5 Gy (LV5) was considered the best for RP estimation than the DVH model [155]. By the combination use of cone-beam CT radiomics features (NGTDM25: Contras and others) and two pretreatment CT radiomics features (SHAPE: Mass and SHAPE: Orientation), the prediction specificity of lung toxicity was further improved from 80.77% to 84.62% after SBRT in stage I NSCLC patients [156]. Therefore, it may contribute to the discovery of more convenient methods to predict RILI for patients with the development of noninvasive molecular imaging.

## 5. Predicting Models of Radiation-Induced Lung Injury

The Quantitative Analysis of Normal Tissue Effects in the Clinic (QUANTEC) report is a pioneer model used to estimate the potential RP risk of increasing dose/volume to an organ at risk (OAR) [157]. Based on this foundation, in addition to dosimetric parameters, clinical-related risk factors are incorporated in various models for better recommendation to guide clinical practice; the Appelt model includes six additional patient characteristics apart from the dosimetric parameters, while the Defraene model includes the dyspnea score [158,159]. Moreover, various updated and generalized normal tissue complication probability (NTCP) models are continually being developed from the classical Lyman model, which considers non-dosimetric factors to compensate for the sole consideration of dose-volume responses and a binary (yes/no) toxicity assessment [160,161]. According to the analysis of a large number of publications, all these models were valuable for patients to obtain the most benefit from radiotherapy with acceptable adverse effects. Unfortunately, it remains difficult to establish an accurate model for precisely predicting lung toxicity in individual patients after radiotherapy, and integration modeling with objective biological indicators of lung damage would definitely improve the NTCP values.

Kong et al. first investigated the effect of cytokine variations and clinical factors on the dose-toxicity relationship. For stage I-III NSCLC patients with definitive radiotherapy, they found a multivariable model that could improve the prognostication for risk variation of grade ≥ 3 lung toxicity compared to the mean lung dose (MLD) alone model [162]. Their study further validated that, among the 30 cytokines studied, IL-8 and TGF-β1 were promising predictors. Compared to MLD alone, a model combined with pre-IL-8 level and TGF-β1 2w/pre (2 week/before irradiation) ratio could predict the risk of grade ≥ 2 toxicity more accurately [86]. For NSCLC patients from four prospective clinical trials, another study recently stated a weighted-support vector machine (SVM) classifier, which integrated the inflammatory cytokine CCL4 with clinical variables to predict grade ≥ 2 fibrosis using machine learning. This model has shown predictive value for fibrosis estimation to a certain extent, but still requires validation with larger sample sizes [85,163]. As mentioned above, in addition to clinical parameters and biological variations, the genetic background of SNPs was closely associated with the individualized sensitivity of lung toxicity. After analyzing genetic variants in lung cancer patients, Liu et al. proposed that the combination of RP-related SNPs, including *PAI-1*, *TGFβ1*, *ITGB6*, *PI3CA*, *AKT2*, and *MMP1* could achieve a more accurate prediction of grade ≥ 3 RP, although extensive trials are needed to confirm this model [164].

## 6. Conclusions and Perspective

Lung complications are the main dose-limiting toxicities in cancer patients after thoracic radiotherapy [149]. Serious radiation pneumonitis with infection usually requires longer hospital stays and higher hospitalization costs, which increases personal and societal healthcare burdens. Pulmonary fibrosis is considered an irreversible pathological process that can lead to dyspnea, impaired lung function, or respiratory failure in patients [165]. In addition, no accurate predictors of lung toxicities exist in the clinic, and few effective drugs are currently available to treat lung fibrosis. Thus, effective early predictors certainly benefit patients with regard to better prevention and control of complications [166]. Based on the main mechanisms of RILI described in Figure 1, this review identified some potential biomarkers of RP and PF, including cell population variations, cytokines, proteins, miRNA expression, and gene characteristics, as shown in Table 1 and Table 2. Several prediction models for RILI have also been discussed in this review. Although no ideal and reliable indicators or risk models are available in clinical practice, encouraging research results are emerging. Many retrospective and prospective studies are currently being conducted, and the molecular mechanisms of RILI are becoming much clearer. Novel integrated models of individualized genetic susceptibility, clinical background parameters, and biological variation are on the horizon illustrated in Figure 2.

## Figures and Tables

**Figure 1 biomedicines-09-01181-f001:**
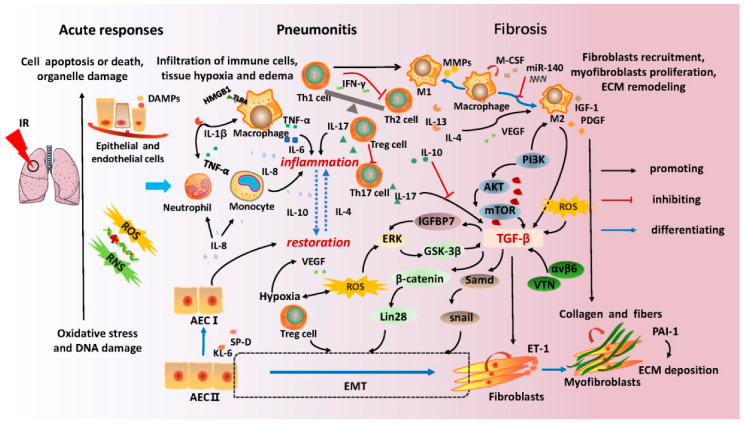
Illustration of the main mechanisms of radiation-induced lung injury. Radiation-induced lung injury contains three inseparable phases along with time: (1) acute radiation responses; (2) radiation pneumonitis; and (3) pulmonary fibrosis. Once exposed to ionizing radiation, cell damage occurs in the alveolar and vascular epithelium. Subsequently, inflammatory cells such as macrophages, neutrophils, monocytes, and lymphocytes are recruited to the damaged tissue with the upregulation of chemotactic factors and adhesion molecules. Cytokines, including TNF-α, IL-1β, IL-6, IL-8, IL-4, and VEGF, are secreted and released to promote inflammation or tissue restoration concurrently. Radiation pneumonitis occurs due to an imbalance between restoration and inflammation. Radiation-induced ROS stimulates downstream signaling via the ERK/GSK-3β/snail axis. Increased GSK-3β then activates TGF-β and leads to an increase in β-catenin levels, which maintains stemness of type II AEC and promotes its differentiation into fibroblasts. Additionally, IGFBP7 is enhanced by TGF-β and is involved in the EMT of AECs through the ERK signaling pathway. Several other molecules, such as αvβ6 integrin and VTN, and signaling pathways, such as the PI3K/AKT/mTOR axis, can also promote the expression of TGF-β. Abundant TGF-β induces fibroblast proliferation and differentiation into myofibroblasts, and promotes collagen synthesis. Excessive deposition of ECM eventually results in pulmonary fibrosis. **Abbreviations:** IR: Ionizing radiation; RNS: Reactive nitrogen species; ROS: Reactive oxygen species; DAMPs: Damage-associated molecular patterns; HMGB1: High-mobility group box 1; TNF: Tumor necrosis factor; IL: Interleukin; TGF: Transforming growth factor; INF: Interferon; IGF: Insulin-like growth factor; AEC: Alveolar epithelial cells; M1: Classic activated macrophages; M2: Alternatively activated macrophages; MMP: Matrix metalloproteinases; M-CSF: Macrophage colony-stimulating factor; PDGF: Platelet-derived growth factor; VEGF: Vascular endothelial growth factor; IGFBP7: Insulin-like growth factor binding protein 7; VTN: Vitronectin; αvβ6: αvβ6 integrin; ERK: Extracellular regulated protein kinases; GSK-3β: Glycogen synthase kinase-3 beta; ECM: extracellular matrix; EMT: Epithelial-mesenchymal transformation; ET-1: Endothelin-1; PAI-1: Plasminogen activator inhibitor-1; KL-6: Krebs von den Lungen-6; SP-D: surfactant protein D.

**Figure 2 biomedicines-09-01181-f002:**
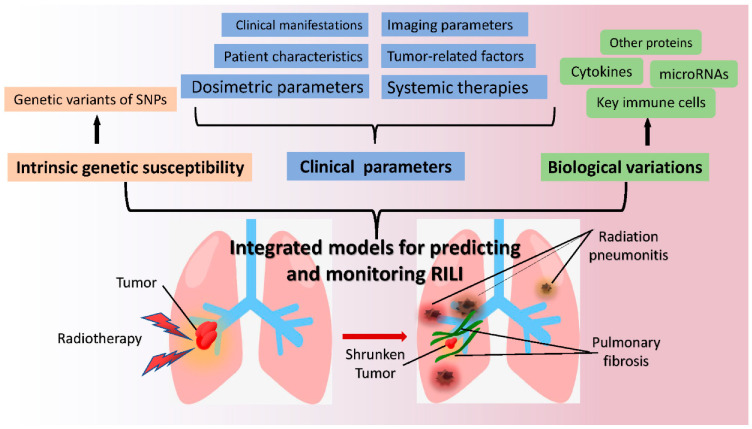
Illustration of the integrated model for predicting and monitoring RILI.

**Table 1 biomedicines-09-01181-t001:** Potential biomarkers for monitoring radiation-induced lung injury.

Categories	Biomarkers	Function	Correlation Research	Ref.
Immune cells	T-cell subsets	T cells determine the specificity of immune responses in tissue inflammation, autoimmunity and host defense	Decreased lymphocytes predicted the severity of RP in LC patients	[72,73,74,75]
	NLR	NLR is an important biomarker of inflammatory status and disease exacerbation	Higher NLR in NSCLC patients with radiological RP predicted the development of symptomatic RP	[76,77]
Inflammation-related factors	IL-1β	IL-1β promotes the recruitment of inflammatory cells and the release of chemokines	IL-1β level was a significant initiator of RILI both in vivo and in vitro studies	[78,79]
	IL-6	IL-6 regulates cells proliferation and differentiation, hematopoiesis, angiogenesis and immune reactions	IL-6 level was a potential monitor for RILI development clinically	[80,81,82]
	IL-10	IL-10 is an anti-inflammatory cytokine by blocking the production of pro-inflammatory cytokines and inhibiting the capabilities of antigen-presenting cells	IL-10 level was low throughout the irradiation period in RP patients, various IL-10 levels monitored different RP scales	[83]
	CRP	CRP is an acute phase inflammatory protein and elevated after injury, infection or inflammation	CRP level in plasma was a potential predictor for RILI development in LC patients	[72]
	IL-8	IL-8 has an anti-inflammatory effect and mediates pulmonary fibrosis	Lower pre-IL-8 level predicted higher risk of grade 2 RILI in LC patients	[84,85,86]
Fibrosis-related factors	TGF-β	TGF-β promotes the differentiation of fibroblasts into myofibroblasts and synthesis of ECM proteins, and reduces collagen degradation, leading to lung fibrosis	Higher TGF-β1 in plasma monitored symptomatic RILI both in vivo and in vitro studies	[51,86]
	ET-1	ET-1 inhibits the proliferation and migration of endothelial cells and promotes ECM production	ET-1 monitored the dynamic changes of PF in mice	[87,88]
	KL-6	KL-6 has chemotactic and anti-apoptotic effects on fibroblasts, leading to lung fibrosis	Increased KL-6 level monitored PF activity and predicted RP severity in patients	[89,90,91]
	PAI-1	PAI-1 inhibits the plasmin system through blocking fibrinolysis and degradation of the ECM	PAI-1 level predicted PF development in patients	[92,93,94]
Chemokines	CCL2/MCP-1	CCL2, also called MCP-1, is a potent chemokine for monocytes	Lower CCL2 level monitored patients with grade 2 RP	[81,86,95]
Other proteins	IFN-γ	IFN-γ is a pleiotropic cytokine with antitumor, antiviral, antibacterial, pro-inflammatory and antifibrotic properties	IFN-γ level indicated the ability to attenuate fibrosis formation in patients	[93,96]
	SP-D	SP-D works in host defense and regulates immune responses and lung phospholipid levels	Elevated SP-D is a sensitive biomarker for early RILI prediction both in patients and murine models	[97,98,99]
miRNAs	miR-21	BMSCs inhibit the pro-inflammatory pathway of macrophage 1 in a miR-21 dependent manner	miR-21 over-expressed in BMSCs significantly alleviated alveolitis in RILI rats	[100]
	miR-140	miR-140 protects lung tissue from fibrosis through blocking TGF-β1 signaling and inhibiting myofibroblast differentiation	Loss of miR-140 in the lung tissue is a key risk factor for PF murine	[101]

**Abbreviations**: RILI: Radiation-induced lung injury; RP: Radiation pneumonitis; LC: Lung cancer; NSCLC: Non-small-cell lung cancer; NLR: Neutrophil-lymphocyte ratio; TGF: Transforming growth factors; IL: Interleukins; CRP: C-reactive protein; PTX3: Pentraxins 3; ET-1: Endothelin-1; KL-6: Krebs von den Lungen-6; IFN: Interferons; ECM: Extracellular matrix; CCL2: Chemokine C-C motif ligand 2; MCP-1: Monocyte chemoattractant protein-1; PAI-1: Plasminogen activator inhibitor-1; EMT: Epithelial-mesenchymal transition; SP-D: surfactant protein D; PF: Pulmonary fibrosis; BMSCs: Bone marrow mesenchymal stem cells.

**Table 2 biomedicines-09-01181-t002:** SNPs in genes associated with radiation-induced lung injury.

SNPs	Year of Publication	Gene Function	Correlation Research	Ref.
*TOPBP1*: rs1051772	2016	DNA repair	decreased risk of RP in NSCLC patients	[102]
*MTHFR*: rs1801131	2017	DNA repair	decreased risk of grade ≥ 2 RP in esophageal squamous cell carcinoma patients	[103]
*NEIL1:* rs4462560	2021	DNA repair	decreased risk of grade ≥ 2 RP in LC patients	[104]
*NEIL1:* rs7402844	2021	DNA repair	higher risk of grade ≥ 2 RP in LC patients	[104]
*PI3CA*: rs9838117 *AKT2*: rs33933140, rs11880261	2016	Inflammation	higher risk of grade ≥ 3 RP in LC patients	[105]
*IL4*: rs2243250	2019	Inflammation	higher risk of grade ≥ 3 RP in LC patients	[13]
*ATG16L2*: rs10898880	2018	Autophagy	higher risk of RP in NSCLC patients	[106]
*PAI-1:* rs7242	2017	Plasmin system inhibition	higher risk of grade ≥ 3 RP in LC patients	[92]
*ITGB6*: rs4665162	2016	Cell surface adhesion	higher risk of grade ≥ 2 RP in LC patients	[107]
*MMP-1*: rs1144393	2018	Protein degradation	higher risk of grade ≥ 2 RILI in LC patients	[108]
*HIPK2*: rs2030712	2020	Cell apoptosis, proliferation and DNA repair	higher risk of grade ≥ 2 RP in LC patients	[12]

**Abbreviations:** SNPs: Single-nucleotide polymorphisms; RILI: Radiation-induced lung injury; RP: Radiation pneumonitis; LC: Lung cancer; NSCLC: Non-small-cell lung cancer; TOPBP1: Topoisomerase DNA binding II binding protein 1; MTHFR: Methylenetetrahydrofolate reductase; NEIL1: Nei endonuclease VIII-like 1; PI3CA: Phosphatidylinositol-4,5-bisphosphate 3-kinase catalytic subunit alpha; AKT2: AKT serine/threonine kinase 2; IL4: Interleukins 4; ATG16L2: Autophagy related 16 Like 2; PAI-1: Plasminogen activator inhibitor-1; ITGB6: integrin beta6 subunit; MMP-1: Matrix metalloproteinase-1; HIPK2: Homeodomain interacting protein kinase 2.

## Data Availability

Data sharing not applicable.

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
