# Peer review of "Promising Biomarkers of Radiation-Induced Lung Injury: A Review"

_biomedicines, 2021, doi:10.3390/biomedicines9091181_

Round 1

Reviewer 1 Report

Research on RILI is a broad field of research encompassing rational animal model establishment, molecular mechanisms, biomarkers, genetic susceptibility, and predictive clinical parameters. Recently, with the development of cancer treatment methods, the cure rate of cancer patients has increased and the number of long-term survivors has increased. Accordingly, the quality of life after cancer treatment has become very important, and research on RILI is an important research topic in this era.

This study is a review paper that systematically organizes a wide range of RILI-related topics by sub-themes. It seems to be able to provide a good guide for researchers in related fields about what has been done so far and the future direction.

Overall, it's impeccable, with only a few minor tweaks.

Minor revision)

- In line 13, abstract, “Patients with severe radiation pneumonitis (RP) or pulmonary fibrosis (PF) will have a poor prognosis, even when the anti-tumor treatment is effective.” As a point of reference, RILI usually has a negative effect on patients' quality of life, but does not significantly affect prognosis like Survival. Please describe it in such a way that it reduces the quality of life rather than the prognosis.

- In line 106, NF-κß should be modified to NF-κB.  B is a English character, not a Greek alphabet

- On line 143, edit Th2 cells from Th 2 cells

- The expression “hyper” in the expression “a single dose or hyper fraction dose” in line 211 is incorrect. Change this sentence to a single or high-dose per fraction dose or a single dose or hypo-fractional dose

Author Response

  1. In line 13, abstract, “Patients with severe radiation pneumonitis (RP) or pulmonary fibrosis (PF) will have a poor prognosis, even when the anti-tumor treatment is effective.” As a point of reference, RILI usually has a negative effect on patients' quality of life, but does not significantly affect prognosis like Survival. Please describe it in such a way that it reduces the quality of life rather than the prognosis.

Response: The statements of “Patients with severe radiation pneumonitis (RP) or pulmonary fibrosis (PF) will have a poor prognosis, even when the anti-tumor treatment is effective.” was corrected as “Severe radiation pneumonitis (RP) or pulmonary fibrosis (PF) will reduce the quality of life, even when the anti-tumor treatment is effective for patients.”

  1. In line 106, NF-κß should be modified to NF-κB.  B is a English character, not a Greek alphabet

Response: In line 106, NF-κß has be modified to NF-κB.

  1. On line 143, edit Th2 cells from Th 2 cells

Response: In line 143, Th 2 cells has be edited to Th2 cells.

  1. The expression “hyper” in the expression “a single dose or hyper fraction dose” in line 211 is incorrect. Change this sentence to a single or high-dose per fraction dose or a single dose or hypo-fractional dose

Response: The statements of “a single dose or hyper fraction dose” was corrected as “a single dose or hypo-fractional dose”.

Reviewer 2 Report

This review article comprehensively summarized the primary mechanisms and preclinical animal models of RILI, and potential biomarkers for monitoring RILI. In addition, predicting models and future perspectives are well described.

I'd like to add some suggestions as belows:

â‘  Table 1 summarized "potential biomarkers for monitoring (or indicating) RILI". But, some biomarkers seem to be predicting RILI. If possible, categorizing the potential biomarkers into predicting or monitoring might be helpful to readers.

â‘¡ Section 5. described predicting models of RILI. Considering only one figure and two tables are included in this manuscript, adding a simple illustration or figure might be helpful for understanding integrated models predicting RILI.

Reviewer 3 Report

This is a nice review of the biological mechanism and potential biomarkers for prediction of radiation induced pneumonitis. I have a few specific comments below.

Treatment approach is also very key for pneumonitis, with the lung being very sensitive to fractionation. It would also be good to discuss the differences in incidence when comparing 3D-CRT vs IMRT.

Early vascular effects would also be good to discuss, including perfusion changes as a potential early biomarker.

Imaging biomarkers and the use of non-invasive imaging for disease monitoring would also be useful to include.

It would be useful to distinguish more clearly between the clinical vs pre-clinical findings.

It would also be helpful when discussing the predictive models that have been developed, what the patient population was and what was the predictive value of the approach as reported by the respective studies.

Author Response

  1. Treatment approach is also very key for pneumonitis, with the lung being very sensitive to fractionation. It would also be good to discuss the differences in incidence when comparing 3D-CRT vs IMRT.

Response: We have added information about the RP incidence between different treatment modalities (IMRT vs 3DCRT) in the first paragraph of Introduction: “With the use of intensity-modulated radiotherapy (IMRT), a significant reduction of RP risk was observed compared to three-dimensional conformal radiotherapy (3DCRT) [4]. IMRT was associated with less ≥ grade 3 pneumonitis (7.9% v 3.5%, P=.039) in the secondary analysis of the RTOG 0617 clinical trial [5].”

  1. Early vascular effects would also be good to discuss, including perfusion changes as a potential early biomarker.

Response: We have added information about the vascular effects and perfusion changes in the first paragraph of new section “4.5 Imaging Based Biomarkers and others” which have been marked.

  1. Imaging biomarkers and the use of non-invasive imaging for disease monitoring would also be useful to include.

Response: We have added information about the use of non-invasive imaging for disease monitoring in the 2nd paragraph of new section “4.5 Imaging Based Biomarkers and others” which have been marked.

  1. It would be useful to distinguish more clearly between the clinical vs pre-clinical findings.

Response: As advised, we have carefully browsed the full text. Clinical findings are used to describe results from patients in clinical practice (distinguished words like “in patients”, “clinically”, “in human”), while pre-clinical findings are mostly from animal models and experiments (distinguished words like “in mice”, “pre-clinically”, “in rats”). For the convenience of readers, We have distinguished the clinical from pre-clinical findings by adding many critical or identified words in the revised manuscript, especially in “Part 4 Potential Biomarkers for Monitoring Radiation-Induced Lung Injury” and in the column “Correlation research” according to the publications. Please see the details that have been marked.

  1. It would also be helpful when discussing the predictive models that have been developed, what the patient population was and what was the predictive value of the approach as reported by the respective studies.

Response: The patient population and the predictive value of the studies have been detailed in section 5 “Predicting Models of Radiation-Induced Lung Injury” where changes have been marked.